# Motives of Future Elementary School Teachers to Be Physically Active

**DOI:** 10.3390/ijerph17124393

**Published:** 2020-06-18

**Authors:** Miguel Ángel Durán-Vinagre, Sebastián Feu, Susana Sánchez-Herrera, Javier Cubero

**Affiliations:** Faculty of Education, University of Extremadura, Avd. de Elvas S/N, 06006 Badajoz, Spain; miduranv@alumnos.unex.es (M.Á.D.-V.); ssanchez@unex.es (S.S.-H.); jcubero@unex.es (J.C.)

**Keywords:** physical activity, motivation, self-determination index, university students

## Abstract

The aim of this study was to determine the relationship between the motives for engaging in sports activities and the self-determination index (SDI) and how this in turn predicts the intentionality of future elementary school teachers to be physically active. Method: A total of 331 first-year students of the teacher training degree participated, 34.4% men and 65.6% women (M = 20.02; SD = 2.55). They answered the following questionnaires: “Behavioral Regulation in Exercise Questionnaire-3”, “Motives for Physical Activity Measure-Revised” and “Intention to be Physically Active”. Results: Fitness, fun and care of one’s appearance are the motives most valued by university students. A regression analysis (structural equation modeling) revealed that appearance and social motives were negatively related to SDI, although the model clearly predicted the intention to be physically active (*R*^2^ = 0.74). A second model, which positively related the appearance and competence motives with the intention to be active, improved the coefficient of determination (*R*^2^ = 90) and fit index. Conclusions: The motives for engaging in physical activity influence university students in different ways in relation to the SDI and strongly predict the intention to be physically active.

## 1. Introduction

The period of adolescence marks the passage from childhood to adulthood and is described as a time of transition where a series of biological, psychological and social changes occur at great speed [1]. In this sense, adolescence is not only an important period in the development and growth of each person [2], but also a time when healthy habits and lifestyles are created [3]. Physical activity during this stage is a key indicator for the establishment of healthy lifestyle habits that would contribute to decreasing the levels of sedentarism and physical inactivity [4,5,6], as well as stimulating the physical, cognitive and social development of adolescents [7].

It is important to note that in recent decades research on physical activity and health increased and studies have shown that regular physical activity is directly related to improved health and longer life expectancy [8,9,10]. It should also be noted that, both in a national and international context, studies on this topic indicate that the number of university students who are physically active is low [5,11,12,13]. This situation is mainly due to the fact that they prioritize other daily recreational activities and do not appreciate the benefits that physical activity can bring to their physical and mental health [5,12].

In order to explain these data, the concept of physical activity has been linked to other psychological factors, such as self-esteem [14,15], emotional processes [16,17], social climate [18] and, most importantly, to motivation [15,19,20,21]. This last aspect plays an essential role in promoting and engaging in physical activity, since human beings behave according to a series of motivational behaviors when it comes to achieving goals and objectives in any context [22,23,24,25].

Motivation also determines the direction and intensity of behavior when participating in physical and sports activities. Direction is understood as the possible goals that a person manifests when he or she feels attracted to them, while intensity is understood as the amount of commitment and effort that they are willing to employ to achieve those goals [26]. Both the intensity and the direction of the motives are relevant to the continuity of a physical and/or sports activity [27].

Motivation is therefore strongly associated with the reasons for participating in sports activities [28] and the reasons that motivate the university population to be physically active are similar to those presented by the general population [29]. Most studies carried out among young university students show that when engaging in physical activity, they express motives related to pleasurable activities that include some social involvement [30,31], even though this also depends on the profile and characteristics of each person [32]. It is worth noting that the improvement or maintenance of health [33] can also be a motivation for students toward physical activity, which is maintained throughout their university studies and extended over time after completion [34]. Other motives most frequently cited by university students are competition, personal ability and physical fitness associated with body image [35,36,37].

In any case, it is essential to continue studying the motivation for physical activity among future teachers (currently first-year university students), along with their reasons for doing so or not, in order to design and adapt the characteristics of sports activities to meet the needs and specific interests of this segment of the population. Therefore, our objective was to determine the degree to which the motives for engaging in sports activities affected the self-determination index (SDI), as the SDI predicts the intentionality of university students in the elementary education teaching degree to be physically active [38]. Initially, a model was proposed with the following hypotheses: The motives for engaging in physical activity are: enjoyment (enjoyment—SDI, H1); health (fitness—SDI, H2); to create new skills (competence—SDI, H4); to create friendships and meet new people (social—SDI, H3); or to improve one’s image (appearance—SDI, H5). These hypotheses predict the self-determination index (SDI), which in turn predicts the intention to be physically active (H6).

## 2. Materials and Methods

### 2.1. Participants 

The sample was composed of a total of 331 university students in their first year of the teacher training degree, 34.4% men and 65.6% women, with an average age of 20.02 ± 2.55. For a census of 490 new students entering the first year, a sampling error of 3.07% was obtained, representing a 95% confidence level. A non-probability sampling was performed and the date and time of the survey were selected for convenience. The sampling was carried out by distributing the questionnaire among all first-year students present in the classrooms at the time of the survey.

### 2.2. Instruments 

Self-determination index (SDI): Calculating the SDI required measuring motivation towards physical activity in the context of physical exercise. For this purpose, the “Behavioral Regulation in Exercise Questionnaire-3” (the Spanish version BREQ-3) was used [39,40]. The BREQ-3 questionnaire consisted of 23 items grouped into six factors that began with the phrase “I exercise...”. The factors were: intrinsic motivation (four items, such as “Because I think exercise is fun”, with a *α* = 0.94); integrated (four items, such as “Because it is in line with my way of life”, with a *α* = 0.95); identified (three items, such as “Because I value the benefits of physical exercise”, with a *α* = 0.83); introjected (four items, such as “Because I feel guilty when I don’t exercise”, with a *α* = 0.72); external (four items, such as “Because others tell me I should do it”, with a *α* = 0.81); and demotivation (four items, such as “Because I don’t see why I have to do it”, with a *α* = 0.83). Cronbach’s alpha values were adequate (α > 0.70) [41].

The self-determination index was calculated with the following formula [42,43]: (3 × INTRINSIC) + (2 × INTEGRATED) + (1 × IDENTIFIED) − (1 × INTROJECTED) − (2 × EXTERNAL) − (3 × DEMOTIVATION).

Regarding the motives for engaging in physical activity, “Motives for Physical Activity Measure-Revised” (MPAM-R) by Ryan, Hicks and Midgley [44], in its Spanish version [45], was used. This instrument measures the motives for exercising in five factors using 30 items on a Likert scale from 1 to 7. The factors and validity for this study were: the fitness factor, which referred to physical activity as a means to maintain or improve health (e.g., “Because I want to have more energy”); the appearance factor, which referred to physical activity as a means to improve body appearance (e.g., “Because I want to improve my appearance”); the enjoyment factor, which referred to physical activity for pleasure and fun (e.g., “Because I enjoy this activity”); the social factor, which referred to physical activity as a means of establishing and maintaining social relationships (e.g., “Because I like to be with my friends”); and finally, the competence factor, which referred to physical activity for the improvement of skills or because it was a challenge (e.g., “Because I want to develop new skills”). For this study, three items from each factor were used (15 items) from the MPAM-R scale (Figure 1) in order to have between 10 and 20 participants for each parameter estimated in the structural equation model [46]. The reliability of the factors used was optimal [41] (*α* ≥ 0.88).

Measure of the intention to be physically active in the university context (MIFAU) is an adaptation [47] within the university context of the scale “Intention to be Physically Active” [48]. The scale is composed of a single factor with five items that measure the intention of university students to be physically active in the future. Cronbach’s alpha values were adequate (*α* = 0.86).

### 2.3. Procedure

Consent was sought from the institution, the Faculty of Teacher Training and from all participants who completed the questionnaire. They were informed that their participation was voluntary and anonymous, respecting the Organic Law 15/1999 of 13 December on data protection. All participants were treated in accordance with the ethical principles and codes of conduct of the American Psychological Association [49]. Before distributing the questionnaires, the general purpose of the study was explained to the participants. The questionnaires were completed in approximately 15–20 min. At least one researcher was present in the classroom at all times and none of the participants reported any difficulties in completing the instrument.

### 2.4. Statistical Analysis

With the SPSS 23 statistical package (IBM Corp. Released 2012. IBM SPSS Statistics for Windows, Version 23, IBM Corp, Armonk, NY, USA), a preview was carried out to determine the nature of the data; subsequently, a descriptive and correlational analyses of the variables were performed. The AMOS statistical package was used to perform a structural equation model (SEM) with a maximum likelihood estimation method to verify the predictive capacity of the motives for doing physical exercise within the self-determination index. The same was then done to the SDI to verify its predictive capacity towards the intention of being physically active (see Figure 1). The sample size (n > 300) was sufficient to perform the analyses [50]. As the multivariate assumption of normality was not met, a bootstrap procedure (1000 samples) was used with a 95% interval [51]. 

The variables used were the ones observed as motives to exercise (MPAM−R) and the intention to be physically active (MIFAU). The SDI variable was calculated from the factors in the BREQ−3 questionnaire [42]. Several indices were used to assess the adequacy of the model fit [52], such as: comparative fit index (CFI), normed fit index (NFI), Tucker−Lewis index (TLI), standardized root mean squared residual (SRMR) and root mean squared error of approximation (RMSEA). A model fit was considered excellent when the values for CMIN/DF > 1, IFC, NFI, TLI were < 0.95, SRMR < 0.08 and RMSA < 0.06. The model fit was considered acceptable when CMIN/DF > 3, IFC, NFI, TLI were < 0.95, SRMR > 0.08 to < 0.1 and RMSEA > 0.06 to <0.08 [51]. We also calculated the reliability of the questionnaires using Cronbach’s alpha, considering as appropriate factors those > 0.70 [41].

## 3. Results

A preliminary analysis was conducted to evaluate the structure of the scales used. A confirmatory factor analysis (CFA) of the BREQ-3 scale was performed and all 23 items and six factors returned acceptable values (Table 1). The CMIN/DF values between 1 and 3 were excellent. The comparative fit index (CFI), the Tucker-Lewis index (TLI) and the normed fit index (NFI) showed values higher than 0.90. The value of the standardized root mean squared residual (SRMR) and the root mean squared error of approximation (RMSEA) of less than 0.08 were adequate.

The SDI was then calculated using the aforementioned formula, giving an average value of 7.65 ± 5.65. Table 2 shows the descriptions of the variables of the study. Health was the motive that presented the highest average value, followed by fun and the improvement of one’s image. 

The intention to be physically active had significant positive correlations with the motives for physical activity (*p* < 0.01). The self-determination index had the highest correlation with enjoyment, followed by competence and fitness (*p* < 0.01).

### Structural Equation Model

Taking into account that not all motives contribute in the same way to improve self-determination toward physical activity, the first model hypothesized that all motives to engage in physical activity would contribute to improve the SDI, which in turn would be directly and positively related to the intention to be physically active in the future (MIFAU questionnaire) (Figure 1). After using the rates of change progressively (relating the covariances), it was concluded that the model fit the data adequately: CMIN = 490.65; DF = 170; CMIN/DF = 2.88; NFI = 0.92; CFI = 0.95; TLI = 0.93; SMRS = 0.096, RMSEA = 0.076; AIC = 612.66.

It was found that the five dimensions of fitness predicted 80% of the variance of the self-determination index (*R^2^* = 0.80). The regression weights between the motives for physical activity and the SDI were statistically significant. The model showed that both the social motive (β = −0.23; *p* < 0.001) and the appearance motive (β = −0.13; *p* < 0.05) were negatively related to the SDI and they exerted the least effect, along with the competence motive (β = 0.22; *p* < 0.01). The model predicted 74% of the variance of the intention to be physically active (*R^2^* = 0.74).

Another model was hypothesized considering the competence and appearance variables, which were motives related to achieving results in the mid to long term. Appearance was a motive that was oriented towards controlled, introjected motivation [53] (correlation appearance—introjected, r = 0.464; *p* < 0.01) and contributed negatively to the SDI; therefore, both variables were directly related to the intention to be physically active (MIFAU) (Figure 2). The model obtained excellent fit values: CMIN = 365.74; DF = 170; *p* < 0.05; CMIN/DF = 2.151; NFI = 0.94; CFI = 0.97; TLI = 0.97; SRMR = 0.044; RMSEA = 0.059. The regression weights between the enjoyment, fitness, social and SDI motives were statistically significant (*p* < 0.001). The social motive had a negative relationship with the SDI. The regression weights between the motives of competence (β = 0.32; *p* < 0.001) and appearance (β = 0.28; *p* < 0.001), along with the intention to be physically active, were statistically significant. The variables of enjoyment, fitness and social explained 78% of the SDI variance. The model explained 90% of the variance in the intention to be physically active.

## 4. Discussion

Future elementary education teachers should promote healthy habits, both through their knowledge and their personal life habits. Therefore, the purpose of this research was to determine if the relationships between the motives for exercising influenced the self-determination index (SDI), an index calculated from the types of behavior that regulate physical exercise [39]. These behavioral patterns are based on the self-determination theory [54] and its relationship with the intention of first-year university students to be physically active. A structural equation model was proposed to predict how the reasons for physical activity (enjoyment, H1; fitness, H2; competence, H3; social, H4; appearance, H5) predicted the self-determination index (SDI) and the intention to be physically active.

In this study, health was the motive most valued by university students, followed by fun and care of one’s appearance. Health, fun and competition were the motives most referred to by the population in other studies [32,45,55,56].

Appearance is one of the most important motives in our society, especially among young people, although it decreases with age [56]. Other studies have found that women at university were more motivated by appearance than men [38]. However, in a study involving Colombian university students, appearance and competition were found to be the least inspiring motives for physical activity [57].

Regarding the first five hypotheses relating the motives for exercising with the SDI (Figure 1), it was found that enjoyment (H1) was the motive most strongly associated with the self-determination index, followed by fitness (H2) and competence (H3). Fun was considered the most self-determined form of motivation [39] because it was associated with the satisfaction of basic psychological needs [20]. Health contributed less to the model than fun; one explanation for this could be that, as other studies have pointed out, it is associated with the regulation identified as providing an internal value [38], which weighs less in the formula that determines the SDI.

Social relationships (H4) and appearance (H5) were negatively related to SDI. Engaging in physical activity to improve physical appearance was one of the motives related to introjected regulation. This regulation occurs when a person seeks to improve his or her self-esteem and gain the approval of others [38]. With regard to social relationships, the data indicated that along with fun and health, they were the most valued when seeking well-being [32]. These results were not in line with those of other studies, which maintained that the social motive was not the most relevant [38,55,56].

As for the sixth hypothesis, model 1 showed that the SDI predicted the intention to be physically active with 74% of variance. The more self-determined university students were more intrinsically motivated, integrated and identified a greater probability that they will be physically active. However, it is important to remember that appearance, which is related to external regulation, was negatively associated with SDI, as were social relations. Taking into account that the improvement of appearance and competition are mid to long term objectives, and that the care of one’s appearance is a type of introjected regulation [38,53], model 2 was proposed, since those motives seem to improve the possibility of being physically active. The variables associated with the SDI, along with the motives of appearance and competition, contributed to 90% of the variance of being physically active. The fit rates of the second model could be considered excellent [52], improving on those of the first model, and the adjustments made were consistent with the evidence available in the literature.

## 5. Conclusions

This research focused on understanding how the motives underlying physical exercise are related to the self-determination index and how these also predict the intention to be physically active in first-year students of a teacher training degree in a southwestern province of Spain. The models analyzed indicated that fun was the main factor that predicted the intention to be physically active but competition, appearance and health also exerted some influence, although to a lesser extent. On the other hand, health was the motive that mattered most to students, followed by fun and appearance. It would therefore be necessary to carry out campaigns among new university students aimed at promoting health and personal image, along with a significant amount of fun, in order to generate habits of physical activity in the future. One of the limitations of the study is due to the personal circumstances of the students, many of them traveling from other towns and cities, far from where the university is located. In addition, their perceptions may have been influenced by their personal, social and economic conditions. The aim of a future research line will be to compare the motivations for physical activity not only of first-year and last-year students of an undergraduate program but also to include last-year high school students, thereby increasing the sample size in the study populations. Additionally, it might be necessary to carry out comparative studies in different countries and to include students who participate in international exchange programs.

## Figures and Tables

**Figure 1 ijerph-17-04393-f001:**
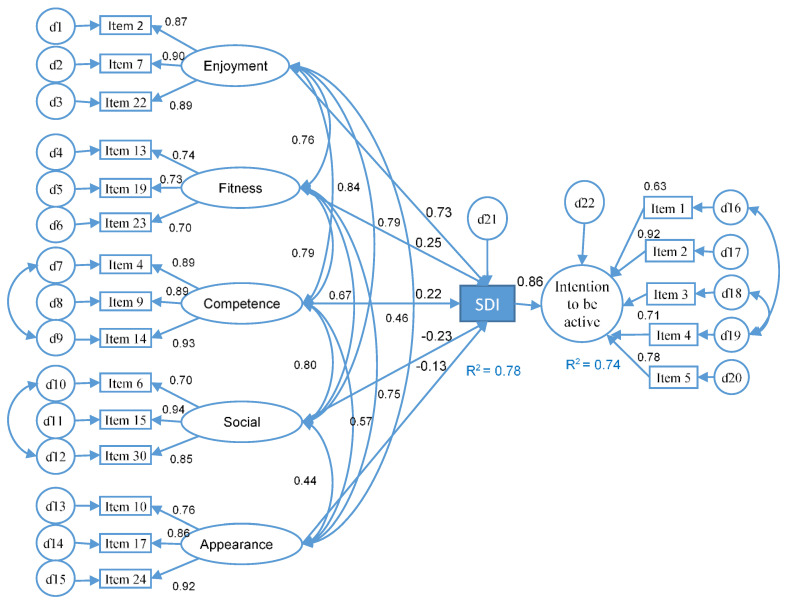
Structural equation model 1 in university students.

**Figure 2 ijerph-17-04393-f002:**
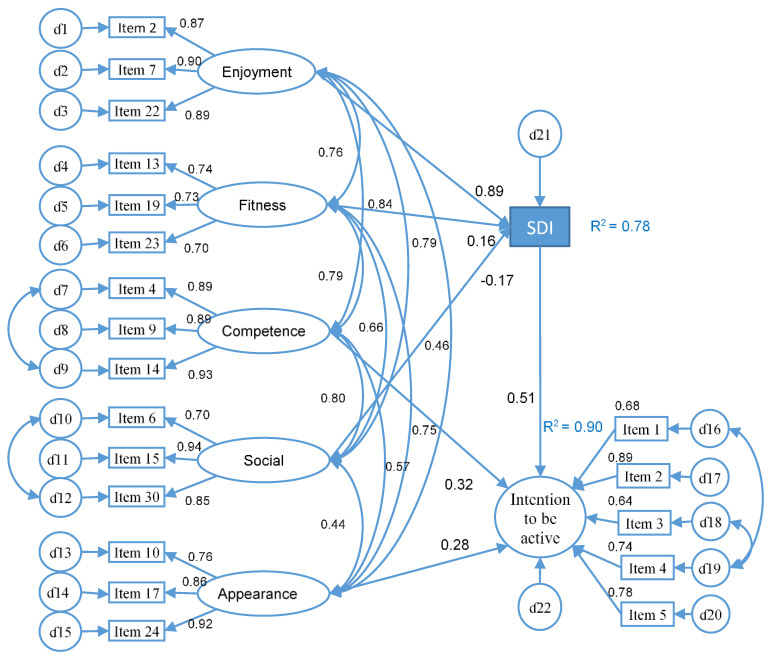
Structural equation model 2 in university students.

**Table 1 ijerph-17-04393-t001:** Confirmatory factor analysis of all the scales.

Scale	CMIN/DF	CFI	NFI	TLI	SRMR	RMSEA
BREQ-3	2.42	0.95	0.92	0.94	0.069	0.066
MPAM-R	1.17	0.98	0.97	0.98	0.031	0.046
MIFAU	0.27	1.00	0.99	1.00	0.008	0.000

**Table 2 ijerph-17-04393-t002:** Descriptive analysis and correlations of the variables.

Factors	Enjoyment	Fitness	Appearance	Social	Competence	MIFAU	SDI
Enjoyment	-	-	-	-	-	-	-
Fitness	0.613 **	-	-	-	-	-	-
Appearance	0.313 **	0.577 **	-	-	-	-	-
Social	0.670 **	0.525 **	0.310 **	-	-	-	-
Competence	0.743 **	0.656 **	0.446 **	0.680 **	-	-	-
MIFAU	0.737 **	0.694 **	0.505 **	0.562 **	0.759 **	-	-
SDI	0.838 **	0.642 **	0.306 **	0.557 **	0.727 **	0.765 **	-
*α*	0.91	0.89	0.88	0.89	0.92	0.86	-
M	4.86	5.38	4.86	4.32	4.41	3.68	10.55
SD	1.72	1.47	1.66	1.73	1.73	1.04	8.74
Skew	−0.71	−1.31	−0.74	−0.45	−0.43	−0.77	−0.42
Kurt	−0.398	1.53	−0.13	−0.78	−0.61	−0.19	−0.74

** *p* < 0.01. M: Mean. SD: Standard Deviation. Skew: Skewness. Kurt: Kurtosis.

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
