# Peer review of "Motives of Future Elementary School Teachers to Be Physically Active"

_ijerph, 2020, doi:10.3390/ijerph17124393_

Round 1
Reviewer 1 Report
Dear authors,
I applaud you for writing this very interesting study about the association between the motives for engaging in sport activities and the self-determination index, along with the ability to predict the level of intention of doing it in the future. You found very strong associations (R2=74%) for appearance and social motives and the appearance and competence motives, which increase the coefficient of determination (R2=90%).
I only have one minor comment or suggestion.
In the 'Study participants' section, I would like to see sample size calculation performed and more detailed information about the type of sampling and the way how it was performed (in the classroom, outside).
Author Response
We thank the reviewer for the useful feedback and suggestions. In the new version of the manuscript, we have reported the sampling error and the type of sampling. The text indicates that the entire census was taken in the classrooms on the day of the survey.
“The sample is composed of a total of 331 university students in their first year of the teacher training degree, 34.4% men and 65.6% women, with an average age of 20.02±2.55. For a census of 490 new students entering the first year, a sampling error of 3.07% was obtained, representing a 95% confidence level. A non-probability sampling was performed, and the date and time of the survey were selected for convenience. The sampling was carried out by distributing the questionnaire among all first-year students present in the classrooms at the time of the survey.”
Reviewer 2 Report
I have only few remarks:
p. 2 line 62-64 - please insert citation for you statement that "the SDI predicts the intentionality..."
p. 2 line 85-86 - please change the brackets for citations.
I had trouble with identifiying right citation and orientation as in the text author used brackets with number but the references are without numbers!
In discussion (p. 6 line 201-202) you mention the gender differences but in your study, you did not differenciate between gender. This factor is quite important, so please can you add justification why you did not study gender differences?
As you measured first year students it would be nice to measure undergraduate students in the last year of their study.
Author Response
We thank the reviewer for the useful comments and suggestions. We have made the following changes:
- P.2 line 62-64: We inserted the following quote
Ozcorta EJF, Almagro BJ, López PS. Predicting intention to remain physically active in university students Cuadernos de Psicología del Deporte. 2015;5(1):275-84.
- P. 2 line 85-86: Parentheses have been changed to square brackets.
- Line 210 shows an error in the translation. As indicated by the reviewer, a gender analysis has not been performed. The sentence was intended to contrast the results with another study where women scored higher than men on the appearance variable. Please see below for the change introduced:
“Appearance is one of the most important motives in our society, and especially among young people, although it decreases with age [55]. Other studies have found that women at university were more motivated by appearance than men [37]. However, in a study involving Colombian university students, appearance and competition were found to be the least inspiring motives for physical activity [56].”
- Following your suggestion, we included the following sentence as a possible future research line:
“The aim of a future research line will be to compare the motivations for physical activity not only of first-year and last-year students of an undergraduate program, but also to include last-year high school students, thereby increasing the sample size in the study populations. Additionally, it might be necessary to carry out comparative studies in different countries, and to include students who participate in international exchange programs.”
Reviewer 3 Report
This study provides an interesting view of what are the motivations, intrinsic and extrinsic, that encourage people to do physical activity, and specifically those that encourage elementary school teachers to involve their pupils in sports activities. The use of two models has been remarkable, since the different variables used in the study are able to influence themselves. Remarkable is also the choice of performing the research on university students, because It’s essential to understand the motivation for physical activity among future teachers in order to design and adapt the sports activities to meet the needs and specific interests of this segment of population.
The article is fluent and well-written, providing clear results.
It would be interesting to widen the sample by including more subjects, also from other regions, to analyze if these results are present also in a wider population.
Author Response
We thank the reviewer for the useful feedback on our work. We are working on analyzing the reasons for engaging in sport activities in other university degrees, and we will include in a future analysis the motivations of university students from other countries and university exchange programs:
“The aim of a future research line will be to compare the motivations for physical activity not only of first-year and last-year students of an undergraduate program, but also to include last-year high school students, thereby increasing the sample size in the study populations. Additionally, it might be necessary to carry out comparative studies in different countries, and to include students who participate in international exchange programs.”